# Stratification of Breast Cancer by Integrating Gene Expression Data and Clinical Variables

**DOI:** 10.3390/molecules24030631

**Published:** 2019-02-11

**Authors:** Zongzhen He, Junying Zhang, Xiguo Yuan, Jianing Xi, Zhaowen Liu, Yuanyuan Zhang

**Affiliations:** 1School of Computer Science and Technology, Xidian University, Xi’an 710071, China; hzz2007.good@163.com (Z.H.); xiguoyuan@mail.xidian.edu.cn (X.Y.); jnxi@xidian.edu.cn (J.X.); 2Psychiatric and Neurodevelopmental Genetics Unit, Center for Genomic Medicine, Massachusetts General Hospital, Boston, MA 02114, USA; liukaka928@126.com; 3School of Computer Engineering, Qingdao University of Technology, Qingdao 266033, China; yyzhang1217@163.com

**Keywords:** gene expression, clinical variables, stratification, mRMR, clustering, luminal-A, luminal-B

## Abstract

Breast cancer is a heterogeneous disease. Although gene expression profiling has led to the definition of several subtypes of breast cancer, the precise discovery of the subtypes remains a challenge. Clinical data is another promising source. In this study, clinical variables are utilized and integrated to gene expressions for the stratification of breast cancer. We adopt two phases: gene selection and clustering, where the integration is in the gene selection phase; only genes whose expressions are most relevant to each clinical variable and least redundant among themselves are selected for further clustering. In practice, we simply utilize maximum relevance minimum redundancy (mRMR) for gene selection and *k*-means for clustering. We compare the results of our method with those of two commonly used only expression-based breast cancer stratification methods: prediction analysis of microarray 50 (PAM50) and highest variability (HV). The result is that our method outperforms them in identifying subtypes significantly associated with five-year survival and recurrence time. Specifically, our method identified recurrence-associated breast cancer subtypes that were not identified by PAM50 and HV. Additionally, our analysis discovered three survival-associated luminal-A subgroups and two survival-associated luminal-B subgroups. The study indicates that screening clinically relevant gene expressions yields improved breast cancer stratification.

## 1. Introduction

Breast cancer is a heterogeneous disease that comprises distinct subtypes reflecting different biological mechanisms and overall survival or recurrence rates [1]. In an organ, different tumor subtypes are considered different types of cancers in the design of treatment [2]. The stratification of cancer patients into subtypes for targeted therapy is one of the goals of breast cancer precision medicine. The subtypes significantly associated with survival or recurrence are referred to as the meaningful subtypes that predict clinical patient outcomes [3]. Unsupervised methods have been used for stratification to get the informative subtypes.

The biological databases currently in existence provide different types of molecular data, such as gene expression. However, they are usually extremely high-dimensional, much larger than the number of samples [4,5]. Dimensionality reduction should be considered before clustering. For example, the number of human genes can reach more than 20,000, but most of them are redundant, and only a few genes are clinically relevant and meaningful for the study of diseases. Therefore, prescreening informative features of high-dimensional variables before clustering can improve the accuracy of subtyping. Existing common dimensionality reduction methods, such as principal component analysis (PCA) [6] and nonnegative matrix factorization (NMF) [7], can extract crucial information from the original variables instead of directly from the original variables; however, it is difficult for clinicians to interpret these extracted features for further use such as drug targets or patient risk prediction [4]. Overall, feature selection is more suitable than feature extraction for discovering and interpreting the biological mechanism of cancer subtypes.

Gene expression profiling has led to the definition of several subtypes of breast cancer. A state-of-the-art breast cancer classifier is PAM50, which maps a tumor sample to one of five subtypes based on the gene expression pattern of 50 selected genes obtained through microarray and quantitative reverse transcriptase PCR (qRT-PCR) [8]. Although PAM50 is more robust than traditional classification systems [9,10,11] that rely on only a few biomarkers, the separation between luminal-A and luminal-B by the various predictors is not consistent, suggesting that these molecular subtypes may not represent distinct coherent sample groups [12]. Choosing expressions with the highest variability is another recently commonly used gene feature selection method [13,14,15] in cancer subtype discovery. Dvir Netanely et al. discovered breast subtypes with better prognostic power than PAM50 using the top 2000 genes with the highest variability [15] using a method referred to as HV in this paper. However, they only clustered tumors based on molecular data, ignoring the value of clinical variables in breast tumor stratification such as estrogen receptor (ER) and histological type.

Clinical data are another crucial source for cancer samples in addition to molecular data. They have not been integrated for cancer stratification but have successfully contributed toward the prognostics of cancer. For example, ER, progesterone receptor (PR), HER2 protein levels, and HER2 genomic amplification are important markers that are predictive and prognostic of breast cancer [16,17,18]. Recently, extensive efforts have been made to incorporate molecular information and clinical data for better prognosis [17]. Yuan Y. et al. [19] and Xu X. et al. [20] both found that the integration of molecular data and clinical variables can predict survival more accurately than the use of molecular data alone.

However, the improvement of integration is very limited because of a large degree of redundancy in the clinical and molecular information after directly merging the data [19,20]. Roughly combining without considering the interaction between both sets of data, which may be correlated, increases feature dimension and redundancy [21]. In Reference [22], meaningful depression subtypes were obtained by integrating brain functional connectivity and clinical variables through considering the correlation between both data in the classification of depression.

In this study, we not only integrate clinical variables into gene expressions for breast cancer stratification, but also consider the correlation between them during the integration. We adopt two steps: gene selection and clustering, where the integration is in the gene selection step: only genes whose expressions have the largest relevance with each clinical variable and lowest redundancy themselves are selected. mRMR is developed for the feature selection of microarray data to simultaneously select highly predictive but uncorrelated features [23]. In practice, we simply utilize mRMR for gene selection for each clinical variable and then selected genes are taken together to compose a candidate feature set for further *k*-means clustering. The results showed that the gene expression selected by mRMR contributed to more informative breast cancer subtypes that were significantly relevant to survival and recurrence time than that selected by either PAM50 or HV. In particular, our method identified recurrence-associated subtypes that were not identified by either of them. We also identified three luminal-A and two luminal-B subtypes with significantly different survival times, which are valuable for more precise subtype diagnosis. The workflow is shown in Figure 1.

## 2. Results

### 2.1. Subtypes of Breast Cancer

In this paper, we utilized mRMR to integrate gene expressions and clinical variables. Genes with the largest relevance to each of the clinical variables and lowest redundancy among themselves were selected together as candidate features for clustering. The number of top relevant genes *pool* and the number of finally selected simultaneously non-redundant genes *k* for clinical variables were determined experimentally. We set *k* = [50, 100, 150, 200] and *pool* = [1000, 2000, 3000] in our experiment and chose the optimal parameters *k* = 100 and *pool* = 2000 as the final parameters in our study based on the clustering result. The clustering outcomes under different parameters are given in Appendix A. For the remaining 12 clinical variables, we utilized mRMR to select the top 100 gene features for each clinical variable and combined these gene features as a candidate gene set with 1064 gene expressions. Each gene feature was independently centered and normalized over the 1035 tumors prior to *k*-means clustering.

To evaluate the performance of the mRMR feature selection method in gene selection, we compared our method, which utilized mRMR feature selection and *k*-means clustering, with recently and commonly used gene selection methods HV [15] and classical PAM50 calls. The HV approach selected the top 2000 features (genes) showing the highest variability (largest standard deviation) to execute *k*-means clustering for breast stratification, which is widely used for extraction of the most informative features from gene expression data. Clusters of both mRMR and HV feature-screening methods exhibited almost the same moderate concordance with PAM50 calls (Table 1). We set five as the cluster number to be consistent with downloaded PAM50 calls.

The clustering performances were evaluated by investigating the relationship between subtypes and patient 5-year survival or recurrence time. As shown in Figure 2, the mRMR method outperformed HV and PAM50 in the survival and recurrence analyses, and the subtypes obtained were most significantly associated with survival and recurrence (with the lowest log-rank test *p*-value < 0.05). In particular, our mRMR method identified recurrence-associated breast cancer subtypes that were not identified by HV and PAM50. In addition, heat maps and principal component analysis (PCA) for the RNA-Seq of genes of subtypes based on mRMR and HV models are presented in Appendix A to enhance the results.

To better evaluate the power of mRMR, which selects 1064 genes whose expression is most correlated with clinical variables, we randomly selected the same 1064 gene expressions for *k*-means clustering 1000 times and identified that 430 of the 1000 *p*-values (significance) of association between subtypes and survival were smaller (more significant) than that of mRMR, and none of the 1000 *p*-values (significance) of association between subtype and recurrence was smaller (more significant) than that of mRMR. This indicates that subtypes obtained by mRMR that integrates clinical information were robustly significantly associated with recurrence rate.

As shown in Figure 3, the resulting clusters exhibited moderate concordance with PAM50 calls: most basal, HER2-enriched, and luminal-B samples were assigned into three different clusters (subtypes 4, 3, and 1, respectively), and subtype 4 exhibited the most overexpressed gene pattern. Notably, most luminal-A samples were divided into three different clusters, a homogenous luminal-A cluster (cluster 5) and two clusters composed of a mix of luminal-A and luminal-B samples (clusters 1 and 2), and most luminal-B samples were split into clusters 1 and 2. It should be noted that 37 normal PAM50 samples were divided into cluster 3 (HER2), cluster 4 (basal), and cluster 5 (luminal-A). There are doubts about the normal-like existence of these tumors as a breast cancer subtype, and these tumors can also be classified as triple-negative (basal) [24].

Based on these results, PAM50 luminal-A and luminal-B subtypes did not adequately capture the variability within the luminal samples. Specifically, luminal-A and luminal-B samples could be further divided into more subtle informative subgroups.

### 2.2. Clusters of Luminal Samples Predict Survival and Recurrence Better than PAM50

The clustering results in the above section indicate that the partitioning of luminal-A and luminal-B samples was not reconstructed by unsupervised clustering based on mRMR-selected gene features, which is consistent with Reference [15].

To further discover the variability among luminal samples, 737 PAM50 luminal samples, including 534 luminal-A and 203 luminal-B samples, were divided into two clusters using the other two models (mRMR and HV). Survival and recurrence analysis performed on the two luminal partitions are shown in Figure 4. The figure showed that luminal partitions obtained using mRMR and HV models were both more significantly relevant with survival and recurrence compared with luminal-A and luminal-B samples called by PAM50, which is consistent with Reference [15]. In particular, the gene features selected using mRMR contributed to a clinically relevant partition of the luminal samples that better predicted both survival and recurrence compared with the HV and PAM50 partitions.

### 2.3. Luminal-A Samples Have Three Distinct Classes Predictive of Survival

As the luminal-A samples that were assigned into three major subgroups (Figure 3) exhibited high variability, and two recurrence-relevant luminal-A subgroups have been discovered using the HV method in Reference [15]; therefore, we concentrated on the PAM50 luminal-A class to discover its underlying substructures. The 534 luminal-A samples were reclustered into two and three groups (Figure 5a,b, respectively) for further discovery. Survival and recurrence analyses were performed on these subgroups. We found that the luminal-A subtypes obtained by the mRMR model were significantly associated with the 5-year survival time, while those obtained by the HV method were not predictive of survival under both cluster numbers (Figure 5). It should be noted that luminal-A subtypes obtained using HV had significant relevance with recurrence instead of survival when the cluster number *k* = 2, which is consistent with Reference [15]. In contrast, our subtypes were not significantly relevant with recurrence time, which may be because we only obtained 819 clinically relevant luminal-A gene features using mRMR, which was less than half of the gene number of HV (2000) and uncovered recurrence-related genes.

For mRMR feature selection, luminal-A subtypes with three cluster number were more predictive of survival than that with two cluster number (Figure 5). Three luminal-A subtypes obtained using mRMR (called LumA-R1, LumA-R2, and LumA-R3 in this paper) exhibited better five-year prognostic value compared with HV (Figure 6b,c). In addition, LumA-R1 and LumA-R3 samples exhibited better survival rates than LumA-R2, while genes were overexpressed in LumA-R1 and LumA-R3 samples compared to those in LumA-R2 samples in the RNA-Seq pattern (Figure 6a).

### 2.4. Analysis of Differentially Expressed Genes in Three Luminal-A Subgroups

We further identified the significantly differentially expressed genes in each subtype of luminal-A samples. The top 1000 most differentially expressed genes (see “Materials and Methods”) were selected for LumA-R1, LumA-R2, and LumA-R3. The overlap of the three gene sets was so small that it means these differentially expressed genes were specific for the subtypes (Figure 7).

In addition, the gene enrichment results are shown in Figure 8. The enriched biological processes and pathways of these genes were also distinct for different subtypes. The most significant categories for LumA-R1-specific genes in the enrichment analysis were related to the immune system regulation function, cytokine-cytokine receptor interaction, and T cell receptor signaling pathway, which is consistent with Reference [15]. Genes differentially expressed in the LumA-R2 samples were enriched in protein transport, and LumA-R3-specific genes were related to cell cycle, cell division, and DNA replication. In summary, LumA-R1 and LumA-R3 samples exhibited better survival prognosis, while overexpressed genes were related to the immune system, cell division, and DNA replication.

### 2.5. Luminal-B Samples have Two Distinct Subtypes Predictive of Survival

Luminal-B samples were not further analyzed in Reference [15]. In this study, the 203 luminal-B samples were reclustered into two groups, each utilizing either a mRMR or HV model. Survival analysis performed on the two subgroups revealed that luminal-B partition obtained using mRMR feature selection exhibited more significant association with five-year survival time (Figure 9) than those obtained using HV.

We designated Luminal-B subtypes based on mRMR model as LumB-R1 (*n* = 86) and LumB-R2 (*n* = 117). Most genes were overexpressed in LumB-R1 samples compared to those in LumB-R2 samples, while LumB-R1 samples exhibited better survival than LumB-R2 samples (Figure 10).

### 2.6. Analysis of Differentially Expressed Genes between Two Luminal-B Subgroups

Through analyzing the significantly differentially expressed gene list for LumB-R1 and LumB-R2 samples using significance analysis of microarrays (SAM) and DAVID, we found that the enriched biological processes and pathways of these genes were also distinct for the two Luminal-B subtypes (Figure 11). Genes significantly overexpressed in the LumA-R1 samples were enriched in cell projection morphogenesis, cilium assembly, and regulation of transcription and DNA-templated. The most significant categories for LumB-R2-specific genes in the enrichment analysis were related to regulation of immune system process, defense response, T cell activation, and the KEGG pathways “cytokine–cytokine receptor interaction” and “primary immunodeficiency”. In summary, LumB-R2 samples exhibited worse survival prognosis and overexpressed genes are related to the immune system and T cell receptor compared to LumB-R1 samples who exhibited better survival and overexpressed genes are related to cell projection morphogenesis and cilium assembly.

## 3. Discussion

We integrated gene expressions and clinical variables utilizing the mRMR algorithm to discover informative subtypes of breast cancer. This method can select gene expressions that have maximum relevance with clinical variables and minimum redundancy themselves. Applying mRMR-selected genes and *k*-means clustering, we found that clinically relevant and non-redundant gene expressions can achieve a superior stratification that is significantly relevant to survival and recurrence compared to the PAM50 and HV methods, which use only expression data. In addition, in luminal tumors, we discovered that three luminal-A subgroups and two luminal-B partitions exhibiting distinct gene expression patterns were significantly relevant to survival.

Since there are a large number of available molecular and clinical data for different cancers in TCGA, our method can be also applied to the precise stratification of other types of cancer. As clinically relevant genes were selected using mRMR for each clinical variable respectively and then taken together, there remains a redundancy in our candidate genes. There are two main opportunities to improve our method in future work. On one hand, multi-label feature selection method can be used to select non-redundant features relevant to multiple clinical variables to improve this method. On the other hand, other multi-omics data, such as mutations, might be integrated into this framework to supplement more information from distinct aspects.

## 4. Materials and Methods

### 4.1. Data Preprocessing

The Cancer Genome Atlas (TCGA) RNA-Seq data and clinical information of breast carcinoma were downloaded from the University of California Santa Cruz (UCSC) cancer browser website (https://xenabrowser.net/datapages/) [25]. The RNA-Seq gene expression dataset (Illumina HiSeq 2000 RNA Sequencing platform, level 3 transcription, log2(x + 1) transformed RSEM-normalized count [26]) contained 1218 samples, and the clinical information included 1248 samples, of which 11 male samples, 8 metastatic patients, 30 samples of unknown tissue source, and 113 true normal samples were filtered out. Our analysis used only the 1035 samples whose expression data, clinical information, and PAM50 calls [8] were available, including 183 basal-like, 78 HER2-enriched, 534 luminal-A, 203 luminal-B, and 37 normal-like samples. PAM50 calls were obtained directly from Reference [15]. Flat genes and clinical variables that had the same values on more than 80% samples were discarded [19]. Finally, 18,624 genes and 12 clinical features (not including survival time and recurrence time) were kept for further analysis. Survival time and recurrence time were applied to evaluate the relevance of the discovered subtypes and clinical outcomes. The clinical features used in our analysis are shown in Table 2. Missing data in clinical variables were imputed using the function na.roughfix() in R software (R, version 3.2.2, Bell Laboratories, NJ, USA) [19].

### 4.2. mRMR Feature Selection

The mRMR is a feature selection method that aims to select a subset of features with high relevance to class and low redundancy between the features themselves [27]. In this study, we utilized mRMR to select features that were most highly correlated with each clinical variable and least correlated among themselves from gene expression data containing 1035 patients and 18,624 genes. Then, the most correlated gene expressions of each clinical variable were selected together to compose a candidate feature set for further clustering. The F-statistic and Pearson correlation coefficient were used to calculate the relevance between gene expressions with discrete and continuous clinical variables, respectively. As gene expression is continuous, for each clinical variable, the Pearson correlation coefficient was utilized to calculate the redundancy among these gene features. The number of selected genes was determined by experiments.

Let us denote the gene expression data for the ith gene with N individuals as $g_{i}\in R^{p},i=1,\ldots ,P$, and let us also denote the clinical data for the jth clinical variable with N individuals as $c_{j}\in R^{Q},j=1,\ldots ,Q$. For each clinical variable c mRMR was utilized to search a gene feature subset S with k features $\{g_{i}\}$, which jointly had the maximal dependency (max-relevance) D(S,c) on the target clinical variable c and the minimal redundancy (min-redundancy) R(S).

For the discrete (categorical) clinical variables, max-relevance was used to search gene features satisfying Equation (1), where relevance D(S,c) was calculated using the mean value of all F-statistic values F of individual gene $g_{i}$ with clinical variable c; for continuous clinical variables, max-relevance was used to discover gene features satisfying Equation (2), where relevance D(S,c) is obtained by the mean value of all Pearson correlation coefficient (PCC) values of individual gene $g_{i}$ with clinical variable c:(1)$$\max D(S,c),\;D=\frac{1}{\left|S\right|}\sum_{g_{i}\in S}F(g_{i};c)$$
(2)$$\max D(S,c),\;D=\frac{1}{\left|S\right|}\sum_{g_{i}\in S}PCC(g_{i};c)$$

The features selected according to max-relevance may have rich redundancy, which indicates that the dependency among these features may be large. When two features were highly interdependent, removing one of them would have little effect on the relevance power. Therefore, the min-redundancy constraint could be adopted to select irrelevant features [27] by Equation (3):(3)$$\min R(S),\;R=\frac{1}{{\left|S\right|}^{2}}\sum_{g_{i},g_{j}\in S}PCC(g_{i};g_{j})$$

The mRMR algorithm first selected the first gene feature whose expression had the largest relevance with each clinical variable. Then, it performed a greedy search and added one feature in each iteration based on the MIQ (mutual information quotient) criterion [23]. The second feature should be selected to satisfy that it was different from the first one and that they were irrelevant to each other. The operator ϕ(D,R) combined the two constraints “maximal-relevance” and “min-redundancy” using Equation (4) to optimize *D* and *R* simultaneously:(4)$$\underset{g_{k}}{\max}\phi (D,R),\phi =D/R$$

Because it was too time consuming to select features from all gene expressions, we should guarantee that all selected genes were from the top relevant features [24]. The number of top relevant genes *pool* and the number of finally selected, simultaneously non-redundant genes *k* for each clinical variable were determined by experiments.

After the mRMR feature selection, the most relevant gene expressions of each clinical variable were taken together to compose a candidate feature set for further clustering.

### 4.3. Unsupervised Clustering

Unsupervised clustering of the breast tumor samples was executed based on mRMR-selected features (genes). Each gene feature was independently centered and normalized over the tumors before clustering as described in Reference [15]. The candidate genes for each sample set were reselected to ensure that the features were most suitable for that set, and the *k*-means clustering method was utilized in MATLAB (release 2013b, MathWorks, Natick, MA, USA) with correlation distance and 100 replicates.

### 4.4. Clustering Assignment and Clinical Analysis

In order to compare the agreement of clustering assignment with the PAM50 calls, the adjusted rand index (ARI) [28,29] was calculated for the clustering results. The analysis of clinical outcomes (survival and recurrence time) was carried out by the R “survival” package. Kaplan–Meier survival and recurrence curves [30] of the subtypes were plotted, and *p*-values for the difference in 5-year survival and recurrence time among subtypes were calculated using the log-rank (Mantel–Haenszel) test [31,32].

### 4.5. Identification of Differentially Expressed Genes and Gene Enrichment

The significantly differentially expressed genes of each luminal-A/luminal-B subtype relative to the remaining luminal-A/luminal-B subtypes were identified using the SAM method [33] based on all informative genes (*n* = 18,624). The q-values were estimated using SAM with the Wilcoxon-rank sum statistic and 1000 permutations. Then, the top 1000 significantly differentially expressed genes with median q-values < 0.05 for each subtype were selected for further analysis [15].

Gene biological processes and pathway enrichment analysis were performed for the significantly differentially expressed genes in each subtype using DAVID 6.8 (https://david.ncifcrf.gov/). Only enriched annotation terms whose q-values were lower than 0.05 were retained.

## Figures and Tables

**Figure 1 molecules-24-00631-f001:**
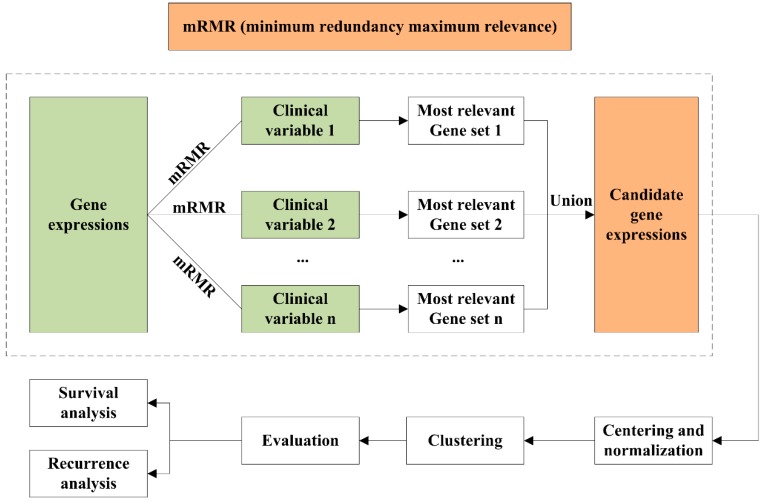
Workflow of integrating gene expression data and clinical variables.

**Figure 2 molecules-24-00631-f002:**
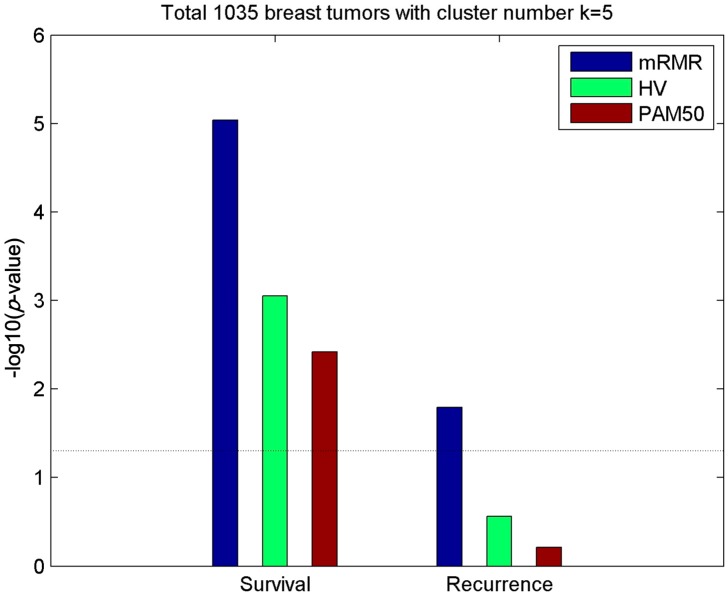
Relevance of survival and recurrence with subtypes of distinct methods. Significance of the association (−log10(*p*-value)) between 5-year survival/recurrence and breast cancer subtypes obtained by mRMR, HV and PAM50, respectively, with cluster number *k* = 5. Dashed lines represent the −log10(*p*-value = 0.05) threshold.

**Figure 3 molecules-24-00631-f003:**
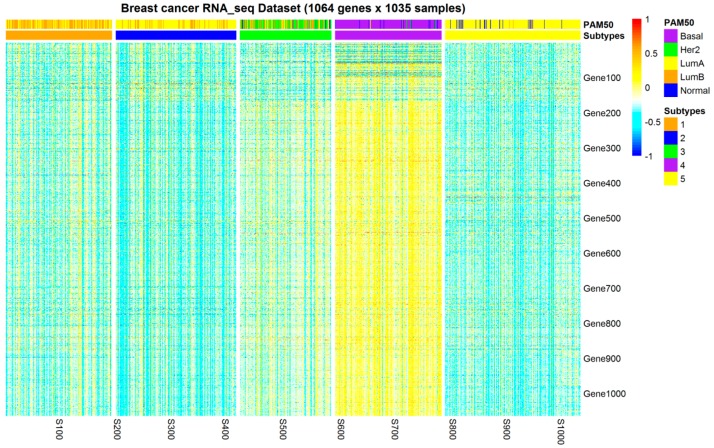
RNA-Seq pattern of clustering subtypes of 1035 breast tumor samples. A partition based on 1064 mRMR-selected gene expressions and the *k*-means algorithm exhibited moderate agreement with PAM50 calls. Notably, luminal-A samples were split into cluster 1, cluster 2, and cluster 5, and luminal-B samples were split into cluster 1 and cluster 2.

**Figure 4 molecules-24-00631-f004:**
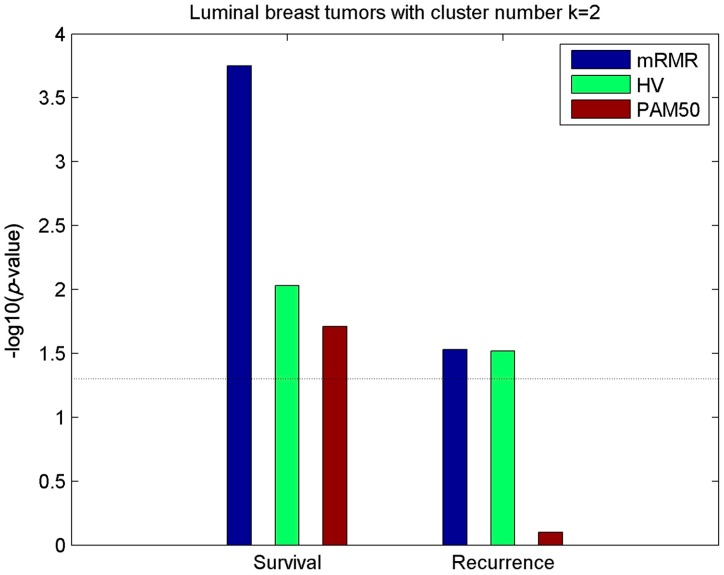
Relevance of survival and recurrence with luminal subtypes of distinct methods. Significance of the association (−log10(*p*-value)) between 5-year survival/recurrence and breast luminal subtypes obtained by mRMR, HV and PAM50 respectively with cluster number *k* = 2. Dashed lines represent the −log10(*p* = 0.05) threshold.

**Figure 5 molecules-24-00631-f005:**
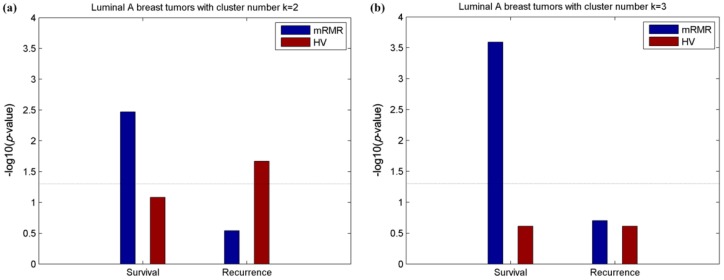
Relevance of survival and recurrence with two and three luminal-A subtypes of distinct methods. Significance of the association (−log10(*p*-value)) between 5-year survival/recurrence and breast luminal-A subtypes obtained by mRMR and HV respectively with cluster numbers *k* = 2 (**a**) and *k* = 3 (**b**). Dashed lines represent the −log10(*p* = 0.05) threshold.

**Figure 6 molecules-24-00631-f006:**
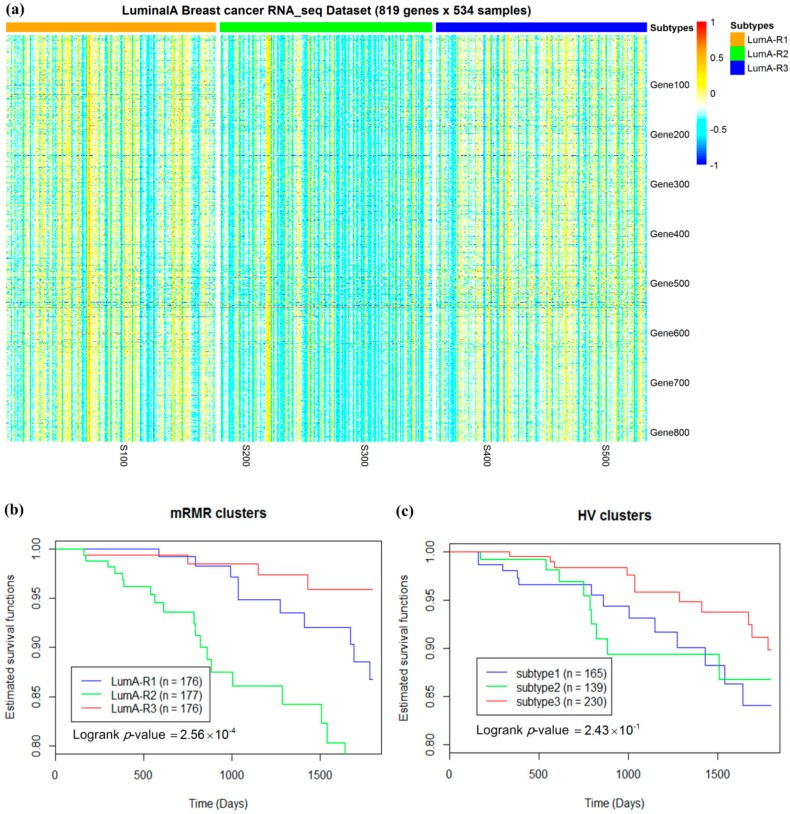
Three subgroups of luminal-A breast samples. (**a**) A partition of 534 luminal-A samples based on 819 mRMR-selected gene expressions and the *k*-means algorithm with cluster number *k* = 3 exhibiting distinct expression profiles. Five-year survival analysis in the three luminal-A subgroups obtained using the mRMR (**b**) and HV (**c**) methods.

**Figure 7 molecules-24-00631-f007:**
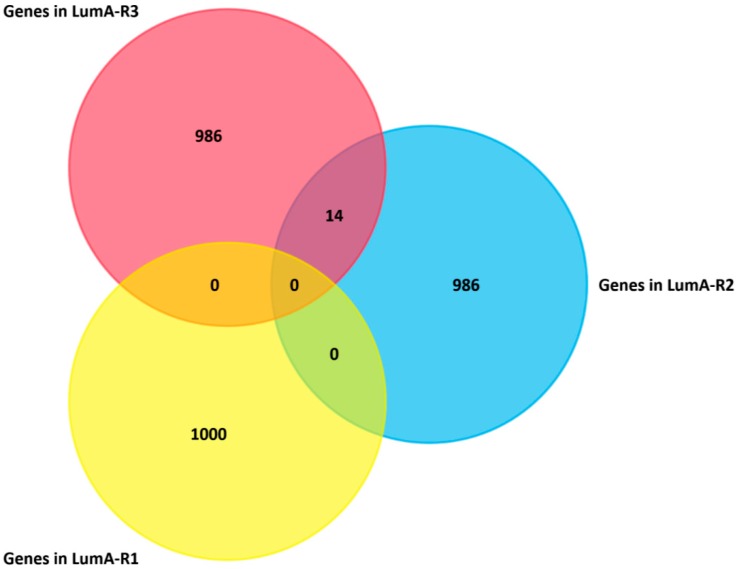
Overlap of differentially expressed genes distinguishing three luminal-A subtypes.

**Figure 8 molecules-24-00631-f008:**
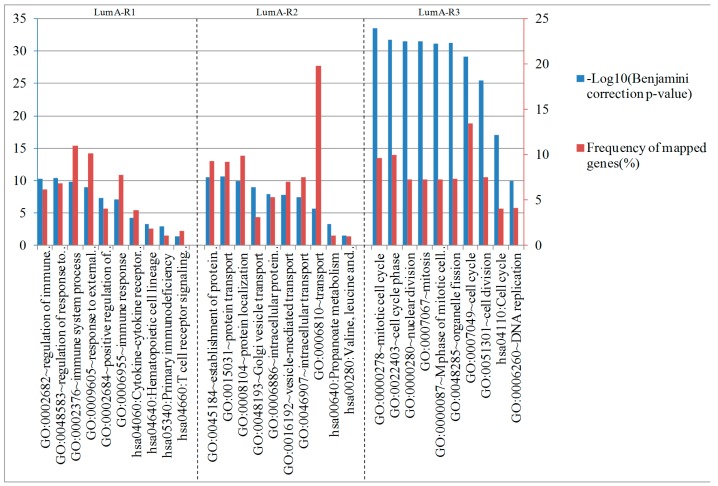
Most significantly enriched biological processes and pathways of the differentially expressed genes in each subtype of luminal-A samples. The blue bars indicate significance with −log10 (Benjamini correction *p*-value), and the red bars indicate frequency of the mapped genes of the corresponding function.

**Figure 9 molecules-24-00631-f009:**
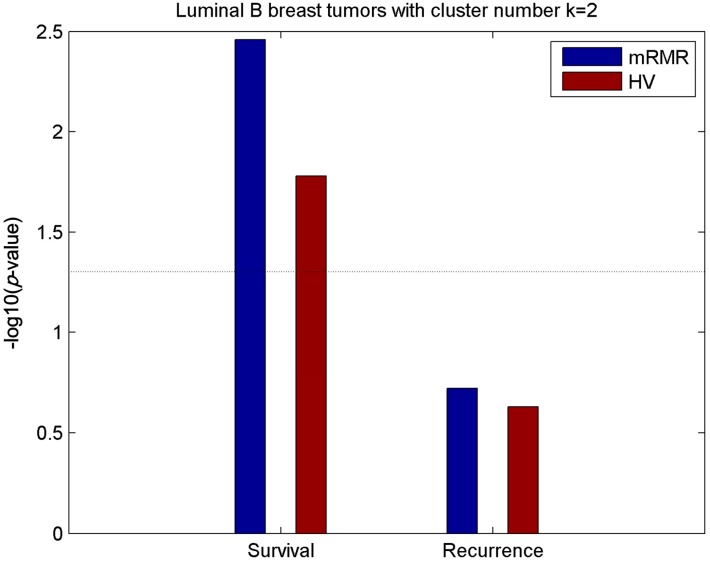
Relevance of survival and recurrence with two luminal-B subtypes of distinct methods. Significance of the association (−log10(*p*-value)) between 5-year survival/recurrence and two luminal-B subtypes obtained using mRMR and HV respectively. Dashed lines represent the −log10(*p* = 0.05) threshold.

**Figure 10 molecules-24-00631-f010:**
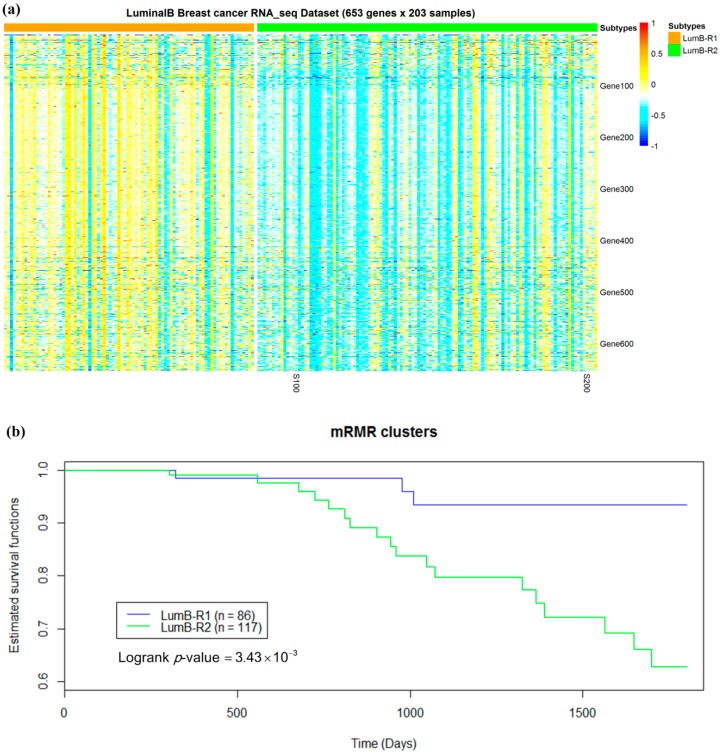
Two subgroups of luminal-B breast samples. (**a**) A partition of 203 luminal-B samples based on 653 mRMR-selected gene expressions and the *k*-means algorithm with cluster number *k* = 2 exhibiting distinct expression profiles. (**b**) Five-year survival analysis in the two luminal-B subgroups obtained by mRMR. Kaplan–Meier survival plots of luminal-B subtypes obtained by the mRMR and *k*-means clustering methods. *p*-values were calculated using the log-rank test.

**Figure 11 molecules-24-00631-f011:**
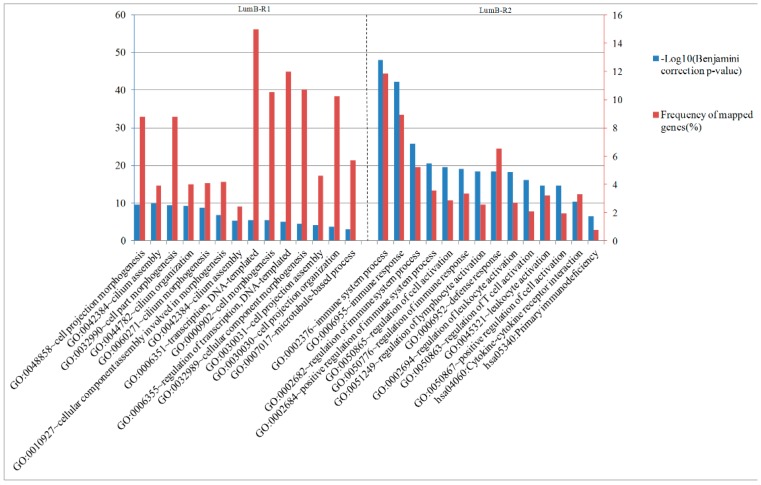
Most significantly enriched biological processes and pathways of the differentially expressed genes in LumB-R1 and LumB-R2 subtypes. The blue bars indicate significance with −log10 (Benjamini correction *p*-value), and the red bars indicate frequency of the mapped genes of the corresponding function.

**Table 1 molecules-24-00631-t001:** Concordances of the clustering results of two feature-screening methods with PAM50 calls.

Screening Method	Number of Features	Adjusted Rand Index with PAM50
mRMR	1064	0.3337
HV	2000	0.3306

**Table 2 molecules-24-00631-t002:** Clinical variables used in our analysis.

Clinical Characteristic	Number of Patients (%)
**Age**	58 years (range 26–90)
**ER.status**	
Positive	761 (74%)
Negative	227 (22%)
Unknown	47 (4%)
**PR.status**	
Positive	658 (64%)
Negative	326 (31%)
Unknown	51 (5%)
**HER2.status**	
Positive	108 (11%)
Negative	649 (63%)
Unknown	278 (26%)
**Stage**	
Stage 1	176 (17%)
Stage 2	589 (55%)
Stage 3	234 (23%)
Stage 4	16 (2%)
Stage 5	13 (2%)
Unknown	7 (1%)
**Histological.type**	
Infiltrating Ductal	753 (73%)
Infiltrating Lobular	182 (17%)
Medullary	5 (1%)
Metaplastic	4 (1%)
Mixed Histology	29 (2%)
Mucinous	16 (2%)
Unknown	46 (4%)
**Pathologic_N**	
N0	491 (47%)
N1	339 (33%)
N2	114 (11%)
N3	71 (7%)
Unknown	20 (2%)
**Pathologic_T**	
T1	269 (26%)
T2	600 (58%)
T3	126 (12%)
T4	36 (3%)
Unknown	4 (1%)
**Cancer_Status**	
Tumor free	883 (85%)
With tumor	115 (11%)
Unknown	37 (4%)
**Lymph_node_count**	10 (range 0–44)
**Tissue_source_site**	10 (range 1–31)
**Year of initial diagnose**	2007 (range 1988–2013)

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
