# Peer review of "Stratification of Breast Cancer by Integrating Gene Expression Data and Clinical Variables"

_molecules, 2019, doi:10.3390/molecules24030631_

Reviewer 1 Report

The publication is of the average scientific merit, not exactly convincing the reader about the "superiority" and/or novelty of the method. It is, though, to be considered for the publication after some changes, major of which are in my opinion necessary in the data presentation:

most of the figures are of the low quality, except for the Kaplan-Mayer curves, which are quite decent (some legends could help). The high-school level diagrams, though, are quite sad.

RNA-seq could be presented by heat maps, or something understandable by reader

Authors should learn some bioinformatics/biocuration and then do good work presenting stuff in table 2... which is at the moment is quite rediculous and do not even represent carcinogenenis, cancer biology, etc. 

English could use some small modifications

Author Response

Response to Reviewer 1 Comments

The publication is of the average scientific merit, not exactly convincing the reader about the "superiority" and/or novelty of the method. It is, though, to be considered for the publication after some changes, major of which are in my opinion necessary in the data presentation:

Point 1: most of the figures are of the low quality, except for the Kaplan-Mayer curves, which are quite decent (some legends could help). The high-school level diagrams, though, are quite sad.

Response 1: Thank you very much for reviewing our manuscript and giving us the chance of revision. All figures in our manuscript have been redrawn and they are all at a sufficiently high resolution (more than 1000 pixels width/height and a resolution of 300 dpi).

Point 2: RNA-seq could be presented by heat maps, or something understandable by reader

Response 2: Thanks for your kind suggestion. We have added heat map of gene expressions for LumB subtypes (Figure 10a) and heat maps of genes for mRMR subtypes and HV subtypes (Figure S1 in Supplement File S2) in the revision. RNA-seq heat maps showed that mRMR subtypes (Figure S1a) exhibit more diverse expression pattern compared to HV subtypes (Figure S1b). We inserted "In addition, heat maps and principal component analysis (PCA) for RNA-Seq of genes of subtypes based on mRMR and HV models were presented in Supplement File S2 to enhance the results." in line 127-129 in revision.

In addition, we have plotted principal component analysis (PCA) for RNA-Seq of genes of subtypes based on mRMR and HV models (Figure S2 in Supplement File S2) to enhance the representation of the results. Through Figure S2, we found that based on three principal components of 1064 gene expressions selected by mRMR, five breast subtypes were more clearly separated by mRMR and HV models compared with PAM50 subtypes (Figure S2a-c). Moreover, subtypes obtained by mRMR model based on 1064 mRMR-selected genes (Figure S2b) were more clearly separated than that obtained by HV model based on top 2000 HV-selected genes (Figure S2d).

Point 3: Authors should learn some bioinformatics/biocuration and then do good work presenting stuff in table 2... which is at the moment is quite rediculous and do not even represent carcinogenenis, cancer biology, etc.

Response 3: Thanks for your valuables suggestion. We are so sorry for our careless mistakes.

We have carefully examined and added the results of GO term and KEGG pathway of the differentially expressed genes in LumB-R1 and LumB-R2 subtypes (Figure 11) in the revision [Line 295-299 in revision] and removed Table 2 in which is only the enrichment results for genes that are significantly overexpressed in LumB-R2. The sentences of results corresponding to Figure 11 were rewritten in revision [Line 262-272 in revision].

Point 4: English could use some small modifications

Response 4: Thanks for your kind comments. We have carefully examined and corrected the grammar and spelling mistakes in our whole revised manuscript.

Reviewer 2 Report

The study by He et al. integrated clinical variables into gene expressions data for breast cancer stratification, and well considered the correlation between them during the integration. The results were well presented and the manuscript was generally well written. I think this MS could be accepted after revision.

While generally well done, I have included below a number of concerns.

Major comment:

Since the authors integrated clinical variables into gene expressions data and identified recurrence-associated breast cancer subtypes that had not been identified by PAM50 and HV, how do we proceed from here then? And can these stratifications be verified by true clinical practice? I suggest the authors give a short outlook on the develops ideas for future research.

Some additional minor comments:

(1)  Line 91: “work flow” should be “workflow”;

(2)  Line 140-141: it seems that besides cluster 4 and cluster 5, the normal samples also can be found in cluster 3.

(3)  Figure 3: why did the authors use the same color to defined “PAM50” and ”subtypes”? It makes the figure a little confused to understand. And what are the colors mean by blue (-1), yellow (0) and red (1) here?

(4)  Line 193: the genes were “overexpressed” or just “induced-expressed”?

(5)  Line 235: the same question as Line 193;

(6)  Line 236: add (Figure 3) to “we found that most genes in the list were overexpressed in LumB-R2 samples compared to those in LumB-R1 samples”;

(7)  Line 238-240: since the authors want to show the results of GO term and KEGG pathway between LumB-R1 and LumB-R2, why they use only one number in Table 2?

(8)  Line 250: the whole part from line 251 to line 269 seems more like “Conclusion” rather than “Discussion”.

Author Response

Response to Reviewer 2 Comments

Point 1: Since the authors integrated clinical variables into gene expressions data and identified recurrence-associated breast cancer subtypes that had not been identified by PAM50 and HV, how do we proceed from here then? And can these stratifications be verified by true clinical practice? I suggest the authors give a short outlook on the develops ideas for future research.

Response 1: Thank you very much for reviewing our manuscript and giving us the chance of revision.

First, we identified recurrence-associated breast cancer subtypes and the recurrence curves of these subtypes are significantly different. From here, for new breast cancer patients, we can assign them to these subtypes by training a classifier with discriminating genes that differentially expressed among the five subtypes. And the prognostic range, especially recurrence time range can be predicted for the new patients and targeted therapy might be executed based on subtype-specific genes.

Second, on one hand, the gene expressions and clinical data used in our study were downloaded from the TCGA database which containing information of true clinical patients. If there are new true clinical patients, we can build a classifier based on our subtypes to predict the subtypes of the new patients. The subtypes of new patients may have significant differences in recurrence time, in theory. On the other hand, we are so sorry that we have no data of new true breast cancer patients for us to verify the stratification.

Third, since there are a large number of available molecular and clinical data for different cancers in TCGA, our method can be also applied on precise stratification of other types of cancer for future research. As clinical relevant genes are selected by mRMR for each clinical variable respectively and then taken together, there remains redundancy in our candidate genes. There are mainly two directions to improve our method in future work. On one hand, multi-label feature selection method can be used to select non-redundant features relevant to multiple clinical variables to improve this method. On the other hand, other multi-omics data, such as mutations, might be integrated into this framework to supplement more information from distinct aspects.

Point 2: Line 91: “work flow” should be “workflow”;

Response 2: Thanks very much for your carefulness. We should say sorry for our careless mistakes. We have carefully examined and corrected the grammar and spelling mistakes in our whole revised manuscript.

Point 3: Line 140-141: it seems that besides cluster 4 and cluster 5, the normal samples also can be found in cluster 3.

Response 3: Thanks for your kind suggestion. We have rewritten these sentences [Line 148 in revision].

Point 4: Figure 3: why did the authors use the same color to defined “PAM50” and ”subtypes”? It makes the figure a little confused to understand. And what are the colors mean by blue (-1), yellow (0) and red (1) here?

Response 4: First, thanks for your kind reminder. In order to match the subtypes with the corresponding PAM50 labels, we use the same color to defined "PAM50" and "subtypes". For example, as most PAM50 basal samples were assigned into subtypes 4, the color of basal and subtype 4 were both set as purple.

Second, gene expressions were centered and normalized before k-means clustering, ranging from -1 to 1. Red (1) represents that the genes are highly expressed and blue (-1) represents that the genes are lowly expressed relative to genes with zero expression values (yellow)

Point 5: Line 193: the genes were “overexpressed” or just “induced-expressed”?

Point 6: Line 235: the same question as Line 193;

Response 5 and 6: Thanks for your comments. The genes were "overexpressed". They were over expressed in one subgroup relative to other subgroup. As these genes were selected using mRMR to keep it relevant with clinical variables and non-redundant themselves in our study. Although, they are not completely independent, we are aimed to identify genes as non-redundant as possible.

Point 7: Line 236: add (Figure 3) to “we found that most genes in the list were overexpressed in LumB-R2 samples compared to those in LumB-R1 samples”;

Response 7: Thank you for your kind suggestion. We should say sorry for our careless mistakes. We have carefully examined and added heat map of gene expressions for LumB subtypes (Figure 10a) in the revision. We found that most genes were overexpressed in LumB-R1 samples compared to those in LumB-R2 samples, while LumB-R1 samples exhibited better survival than LumB-R2 samples. We have rewritten the results in the revision [Line 245-247 in revision] and modified the caption of Figure 10 [Line 256-258 in revision].

Point 8: Line 238-240: since the authors want to show the results of GO term and KEGG pathway between LumB-R1 and LumB-R2, why they use only one number in Table 2?

Response 8: Thank you for your valuable suggestion. We are so sorry for our careless mistakes. We have carefully examined and added the results of GO term and KEGG pathway of the differentially expressed genes in LumB-R1 and LumB-R2 subtypes (Figure 11) in the revision [Line 295-299 in revision] and removed Table 2 in which is only the enrichment results for genes that are significantly overexpressed in LumB-R2. The sentences of results corresponding Figure 11 were rewritten in revision [Line 262-272 in revision].

Point 9: Line 250: the whole part from line 251 to line 269 seems more like “Conclusion” rather than “Discussion”.

Response 9: Thank you for your suggestion. We have rewritten the "Discussion" section in the revision [Line 316-331 in revision].